# A Novel End-to-End Deep Learning Framework for Chip Packaging Defect Detection

**DOI:** 10.3390/s24175837

**Published:** 2024-09-08

**Authors:** Siyi Zhou, Shunhua Yao, Tao Shen, Qingwang Wang

**Affiliations:** 1Faculty of Information Engineering and Automation, Kunming University of Science and Technology, Kunming 650500, China; zhousiyi@stu.kust.edu.cn (S.Z.); shentao@kust.edu.cn (T.S.); 2Yunnan Key Laboratory of Computer Technologies Application, Kunming University of Science and Technology, Kunming 650500, China; 3Wuxi Aiyingna Electromechanical Equipment Co., Ltd., Wuxi 214028, China; shyao_ayn@126.com

**Keywords:** chip packaging defect detection, Vision Mamba, dual-stream decoder, feature correlation, X-ray image segmentation

## Abstract

As semiconductor chip manufacturing technology advances, chip structures are becoming more complex, leading to an increased likelihood of void defects in the solder layer during packaging. However, identifying void defects in packaged chips remains a significant challenge due to the complex chip background, varying defect sizes and shapes, and blurred boundaries between voids and their surroundings. To address these challenges, we present a deep-learning-based framework for void defect segmentation in chip packaging. The framework consists of two main components: a solder region extraction method and a void defect segmentation network. The solder region extraction method includes a lightweight segmentation network and a rotation correction algorithm that eliminates background noise and accurately captures the solder region of the chip. The void defect segmentation network is designed for efficient and accurate defect segmentation. To cope with the variability of void defect shapes and sizes, we propose a Mamba model-based encoder that uses a visual state space module for multi-scale information extraction. In addition, we propose an interactive dual-stream decoder that uses a feature correlation cross gate module to fuse the streams’ features to improve their correlation and produce more accurate void defect segmentation maps. The effectiveness of the framework is evaluated through quantitative and qualitative experiments on our custom X-ray chip dataset. Furthermore, the proposed void defect segmentation framework for chip packaging has been applied to a real factory inspection line, achieving an accuracy of 93.3% in chip qualification.

## 1. Introduction

In recent years, semiconductor chips have played a critical role in various industries, including military and defense, electrical engineering, automotive manufacturing, and healthcare [1]. The ongoing developments in manufacturing have resulted in the miniaturization and increased integration of semiconductor chips, thereby significantly enhancing their computational capabilities [2]. However, these developments have also led to the emergence of more intricate chip architectures, which have in turn given rise to considerable difficulties in the field of high-quality chip packaging. Chip packaging entails the secure attachment of an integrated circuit chip to a substrate, followed by the soldering of a protective cover plate onto it. This process serves to safeguard the chip from physical damage and chemical corrosion while simultaneously ensuring optimal electrical connections and thermal management. The discrepancies in manufacturing techniques and thermal stress between materials can result in the formation of voids within the solder layer. The formation of voids can result in a number of adverse effects, including electrical failures, thermal issues, and impaired interconnectivity. These can have a significant impact on the performance and lifespan of the chip in question [3]. Conventional detection techniques employ X-ray imaging systems to obtain internal chip images, which are then manually analyzed to identify any defects. However, this approach is both labor-intensive and time-consuming. Furthermore, the low contrast of X-ray grayscale images diminishes the efficiency and accuracy of defect identification. It is therefore imperative that advanced and reliable technologies be developed with the aim of accurately and efficiently detecting internal defects.

The advent of machine vision has mitigated some of the constraints associated with manual inspections. Dunderdale et al. [4] utilized a combination of Scale-Invariant Feature Transform (SIFT) descriptors and a random forest classifier for defect detection in photovoltaic modules, achieving a classification accuracy of 91.2%. Li et al. [5] developed a method for classifying wafer maps by integrating supervised SVM classifiers with unsupervised Self-Organizing Map (SOM) clustering, achieving a classification accuracy exceeding 90%. Liu et al. [6] employed a hybrid recognition method integrating mathematical morphology with pattern recognition to analyze the causes and types of Printed Circuit Board (PCB) defects. The authors proceeded to process and binarize the raw color images, subsequently proposing a distortion detection algorithm for the threshold segmentation of PCB defect images. This approach was found to yield high levels of accuracy in detection and to reduce the time required for this process. However, traditional machine vision methods rely on handcrafted features, which are often domain-specific and lack generalizability [7]. Furthermore, manual feature design is time-consuming and labor-intensive, making it challenging to adapt to new products and production lines. With the advent of Industry 4.0 [8], there is an increasing requirement for the development of flexible and adaptable defect detection systems.

In recent years, supervised deep learning methods have been extensively employed in industrial product defect detection [9]. These methods can be classified into three categories based on the level of granularity of the computer vision tasks involved. These categories are image-level classification, region-level object detection, and pixel-level semantic segmentation. The preliminary stage of fundamental image-level classification techniques entails the deployment of Convolutional Neural Networks (CNNs) for the extraction of features from samples, which are subsequently classified. Alvarenga et al. [10] proposed a method for the classification of rail defects based on the analysis of wavelet-transformed eddy current signals using a convolutional neural network. The method demonstrated a classification accuracy of 98% when evaluated in real-world settings. Deng et al. [11] proposed an automated defect verification system combining a fast circuit comparison algorithm with deep neural-network-based classification. This approach achieved high accuracy in PCB defect classification and significantly reduced false positive and false negative rates. Shu et al. [12] introduced the Parallel Spatial Pyramid Pooling Network (PSPP-net), which combines online and offline deep CNN feature extraction streams and uses GPU-integrated features and softmax regression to classify LED chip defects. Batool et al. [13] proposed a Convolutional Neural Network (CNN) approach to address class imbalance through data undersampling, offering a robust solution for wafer defect classification. The method achieved an accuracy of 90.44% in the testing phase. While image-level classification methods are capable of identifying defect types in individual samples, they are not appropriate for images that contain multiple defects. In contrast, region-level object detection techniques are capable of identifying and localizing multiple defects within a single image. Tang et al. [14] used MobileNetV3 as a baseline model and combined it with a dual-domain attention mechanism to propose a lightweight PCB defect detection network designed to efficiently detect small defect features. Dlamini et al. [15] developed a detection system for PCB surface mount technology using a feature pyramid network in conjunction with MobileNetV2. Chen et al. [16] enhanced the YOLOv3 model with DenseNet to improve SMD LED chip detection and optimized it using the Taguchi method. In response to the complex architecture and slow inference of anchor-based target detection algorithms, Chen et al. [17] proposed an anchor-free target detection algorithm (LGCL-CentreNet) for PCB defect detection. A lightweight module was designed with the objective of augmenting the local and global contexts, thereby achieving higher defect detection with lower computational complexity. However, due to the irregularities of defects and the wide variations in shapes and sizes, region-level object detection methods only output bounding boxes. As a result, accurate description of the area and shape of complex defects is difficult. Consequently, research has concentrated on pixel-level segmentation methods, with the objective of achieving precise defect localization. Furthermore, precise defect descriptions facilitate subsequent defect identification and improvements in manufacturing processes. In the field of industrial defect detection, researchers have typically designed models based on basic semantic segmentation networks, including FCN [18], U-Net [19], SegNet [20], and the DeepLab series [21,22,23]. These models have primarily focused on multi-scale and lightweight architectures. Ling et al. [24] developed a deep twin semantic segmentation network for the detection of PCB soldering defects, utilizing similarity metrics derived from the twin network to enhance the accuracy of the detection process. Wu et al. [25] proposed a Mask R-CNN-based deep learning method for PCB solder joint defect segmentation, utilizing ResNet-101 as the backbone network, with an mAP of over 95% in segmentation tasks. Yang et al. [26] introduced a novel non-destructive defect segmentation network, called NDD-Net, which uses an Attention Fusion Block (AFB) and a Residual Dense Connection Convolution Block (RDCCB). When applied to the publicly available GDXray and RSSDs X-ray datasets, NDD-Net outperforms other state-of-the-art segmentation networks in defect localization, particularly in overcoming class imbalance challenges.

Despite the widespread application and significant achievements of deep learning algorithms in industrial defect detection, defect segmentation in X-ray images of chip interiors remains challenging. First, the small size of chips and variations in manual placement often result in captured images containing excessive background information, resulting in unnecessary computational load and reduced detection speed. Second, significant scale variations in defects require further research into effective multi-scale feature representation methods. Finally, the lack of semantic information in X-ray images, low contrast between defects and background, and noise interference often blur defect boundaries. Therefore, achieving efficient and accurate segmentation of void defects in chip X-ray images remains a challenge.

To address these challenges, we propose a deep-learning-based framework for chip packaging void defect segmentation that achieves high detection accuracy and speed. The main contributions of this work are as follows:We propose a lightweight U-Net network with rectangular spatial attention (RSALite-UNet) for segmenting chip cover plates with different rotation angles. This network uses MobileNetV3 as the feature extraction backbone and incorporates a rectangular spatial attention module. This design significantly increases the segmentation speed while improving the segmentation accuracy for chip cover plates.We propose a dual decoder Mamba U-Net (DM-UNet) network for void defect segmentation in chip weld areas. This network uses Vision State Space (VSS) blocks as the core units of the encoder to cope with the diversity of defect shapes and sizes. It also introduces the Feature Correlation Cross Gate (FCCG) module, which effectively integrates boundary-aware features with defect segmentation features in the dual decoders, significantly improving chip void defect segmentation performance.

The source code will be available at https://github.com/Zhousiyi0107/Chip-packaging-defect-detection.git (accessed on 10 August 2024).

## 2. Materials and Methods

### 2.1. X-ray Image Acquisition

The X-ray images of the chips used in this study were obtained using an X-ray inspection machine, as shown in Figure 1a. The machine, an X6600 model manufactured by Seamark ZM, shenzhen, China, consists mainly of five parts: X-ray tube, support system, flat panel detector (FPD), imaging system, and computer control system. The machine supports CNC inspection mode and fast automatic detection over multiple arrays. Figure 1b illustrates the principle of chip X-ray imaging. During image acquisition, an X-ray tube generates a beam of high-energy X-rays. As X-rays pass through the chip, different materials absorb them to varying degrees. Denser materials or those with a higher atomic number (such as metals) absorb more X-rays, while less dense materials (such as ceramics) absorb fewer. Consequently, the intensity of the X-rays changes as they pass through the chip. The FPD receives X-rays from the X-ray tube and converts the X-ray intensity signal into a digital image.

The chip is packaged using a ceramic base, with the packaged chip consisting mainly of a ceramic base, a metal cap, and metal leads. Due to the high absorption of X-rays by metal, the metal cover plate and leads appear as deeper shadows in the X-ray image. In addition, X-rays pass through defective and non-defective areas with different degrees of attenuation, resulting in differences in the intensity of the X-rays reaching the detector, which produces images of different brightness on the computer display. Figure 2 shows an example of a raw X-ray image of a chip taken by an X-ray inspection system at a resolution of 1536 × 1536. Due to the small size of the chip, most of the X-rays reach the detector without passing through it, resulting in a strong X-ray signal. As a result, the acquired X-ray image of the chip contains a significant amount of bright (almost white) background information.

### 2.2. Datasets

The performance of the void defect segmentation network is greatly influenced by both the quantity and quality of the dataset. As production processes have evolved, it has become increasingly difficult and costly to collect large numbers of images of defective packaged chips. In addition, pixel-level annotation is very costly. In this study, we collected 1950 raw X-ray images of chips from a real industrial environment. Due to the presence of irrelevant background information in the original images, we cropped the central 800 × 800 pixel region from the raw X-ray images to create a packaged chip X-ray image dataset, named CIDX-Ray1950. We annotated the chip cover contours at the pixel level on 1350 X-ray images, including 1024 unqualified chips (NG) and 326 qualified chips (OK), for training and testing the chip cover plate segmentation model. Of the 1350 X-ray images, 1024 unqualified chips (NG) were further annotated for chip void defects, and these annotated images were used to train and test the chip package void defect segmentation model. The remaining 600 X-ray images, with equal proportions of defective and non-defective images, were used to further test the overall performance of the proposed chip void defect segmentation framework. Detailed are presented in Table 1.

### 2.3. Chip Cover Plate Segmentation and Extraction

Larger image sizes not only reduce the quality of the image analysis but also affect the processing speed. To minimize the effect of background on chip void defect segmentation, we first used a lightweight RSALite-UNet mesh to segment the chip cover plate region. We then used image processing techniques to rotate the chip cover plate region to a horizontal position and then cropped it to extract the cover plate region.

#### 2.3.1. Chip Cover Plate Segmentation

U-Net, a fully convolutional neural network architecture originally proposed by Ronneberger et al. [19] for biomedical image segmentation, features a symmetric encoder–decoder structure with skip connections that effectively captures image features across scales, making it highly suitable for various segmentation tasks. Because of its effectiveness, we use the U-Net architecture as the basis for our segmentation tasks. However, the high computational demand of U-Net when processing high-resolution images is a significant limitation in industrial applications where real-time processing is essential. To address this, we propose a more efficient model, RSALite-UNet, which aims to improve both the speed and accuracy of chip cover segmentation, as shown in Figure 3. This model uses the lightweight MobileNetV3 [27] for feature extraction, significantly reducing the number of parameters and the computational load while preserving essential feature information. The decoder in RSALite-UNet consists of convolutional blocks and bilinear interpolation for upsampling, where each convolutional block contains two 3×3 convolutional layers, followed by batch normalization (BN) and ReLU activation. To further improve segmentation accuracy, we introduce a rectangular spatial attention module within the skip connections between the encoder and decoder, which allows the decoding network to capture both detailed and global information more effectively.

**Rectangular Spatial Attention Module:** As the network depth increases, spatial information tends to be lost, which is critical for accurate chip cover segmentation. Therefore, we propose a rectangular spatial attention module to address this issue. This module uses global max pooling and average pooling along the channel dimension to capture global context. By concatenating the results of max pooling and average pooling, followed by a convolution, it effectively models regions of interest. Next, a pair of 7 × 1 and 1 × 7 depthwise strip convolutions [28] are used to calibrate the attention map vertically and horizontally. Specifically, the vertical depthwise strip convolution calibrates the shape vertically. The features are then normalized using batch normalization (BN), with non-linearity introduced through ReLU activation. Subsequently, the horizontal depthwise strip convolution calibrates the shape horizontally. This decoupling of convolutions in both directions allows the module to adapt to different placements of the rectangular chip cover. The process of calibrating the rectangular spatial attention map is expressed as follows:(1)$$\xi_{C}\left(\bar{y}\right)=\sigma \left({\mathrm{DWConv}}_{7\times 1}\left(\mathrm{BN}\left(\mathrm{ReLU}\left({\mathrm{DWConv}}_{1\times 7}\left(\bar{y}\right)\right)\right)\right)\right)$$
where ${\mathrm{Conv}}_{7\times 1}$ and ${\mathrm{Conv}}_{1\times 7}$ represent the 7×1 and 1×7 depthwise strip convolutions, respectively, and σ denotes the sigmoid activation function.

To better fuse the rectangular spatial attention features with the input features, we apply a 3 × 3 depthwise convolution to further extract local details from the input features.
(2)$$\xi_{F}\left(x,y\right)={\mathrm{DWConv}}_{3\times 3}\left(x\right)⊙y$$
where ${\mathrm{DWConv}}_{3\times 3}$ represents the 3×3 depthwise convolution, and ⊙ denotes the Hadamard product.

**Loss Function:** Therefore, we use the Weighted Binary Cross-Entropy (WBCE) loss function to guide model training, defined as follows:(3)$$L_{\mathrm{WBCE}}\left(y,\hat{y}\right)=-\left(\alpha y\log\left(\hat{y}\right)+\beta \left(1-y\right)\log\left(1-\hat{y}\right)\right)$$
where α and β are the weights for the positive and negative classes, respectively. Assigning different weights to positive and negative samples improves the network’s ability to learn from minority classes. This approach effectively mitigates the effects of class imbalance between foreground and background, thereby improving the network’s segmentation performance.

#### 2.3.2. Chip Cover Plate Extraction

In X-ray images, the orientation of the chip cover area varies, so it is essential to rotate and correct the area before cropping. The detailed process for extracting the chip cover area is as follows: First, the chip X-ray image is fed into the RSALite-UNet segmentation network to obtain a binary mask. Next, the Canny edge detection algorithm [29] is applied to the binary mask to extract the contours of the chip cover area. The minAreaRect function from the OpenCV library is then used to determine the center point, height, width, and rotation angle of the chip cover area. Based on the center point and rotation angle, an affine transformation is applied to the X-ray image, rotating the chip cover area to a horizontal position. Finally, the chip cover area is cropped from the transformed image using the width and height values returned by the minAreaRect function. The extracted chip cover area is then used as input for the subsequent chip void defect segmentation network.

### 2.4. Chip Void Defect Segmentation

We propose a Dual Decoder Mamba U-Net (DM-UNet) for chip void defect segmentation, the architecture of which is shown in Figure 4. This architecture consists of three key components: a Mamba-based encoder, a feature correlation cross gate module, and a dual-stream decoder. The following sections provide a detailed description of these three key components.

#### 2.4.1. Mamba-Based Encoder

CNNs face challenges in capturing long-range dependencies due to their limited receptive fields, which can lead to inadequate feature extraction and poor segmentation results. On the other hand, transformer-based models are highly effective for global modeling, but their self-attention mechanisms involve quadratic computational complexity, leading to a significant computational burden [30]. In contrast, Vision Mamba [31] maintains linear complexity while effectively modeling long-range dependencies, demonstrating superior accuracy with lower computational and memory overheads in visual tasks. Consequently, we adopt Vision State Space (VSS) blocks as the core units of the encoder. The Vision State Space (VSS) block contains a two-dimensional selective scanning (SS2D) module. SS2D bridges the gap between the ordered nature of one-dimensional selective scanning and the non-sequential structure of two-dimensional visual data by traversing four scan paths. This approach gathers global contextual information from multiple sources and angles, addressing the size variation of chip defects. The SS2D module comprises three key components: the scan expansion operation, the S6 block, and the scan merging operation. For a more comprehensive explanation of SS2D, please refer to [31]. The overall architecture of the VSS block is depicted in Figure 5a. The encoder is organized into five stages. The first stage consists of two successive 3×3 convolution blocks. Each subsequent stage contains VSS blocks for high-level feature extraction and max pooling operations for 2× downsampling. The number of VSS blocks in the second to fifth stages is $\left\{2,2,9,2\right\}$. The feature dimensions after each stage are $D=\left\{16,24,32,48,64\right\}$. Using the feature extraction capabilities of VSS blocks allows the use of fewer feature dimensions, significantly improving computational efficiency.

#### 2.4.2. Feature Correlation Cross Gate Module

In comparison to images captured by conventional cameras, X-ray images of chips exhibit a restricted range of colors and low contrast, which results in indistinct boundaries between defective and non-defective regions. In order to address this issue, we propose the implementation of a Feature Correlation Cross Gate (FCCG) module. The module effectively combines features from the two decoder branches by capitalizing on their correlations, thereby improving the precision of chip void defect segmentation. In particular, as illustrated in Figure 5b, the segmentation decoder features, denoted by $F_{\mathrm{seg}}$, and boundary-aware features, denoted by $F_{\mathrm{edge}}$, are initially concatenated. Subsequently, a convolutional layer with a kernel of size 1×1 fuses these features into a single representation. The fused features, designated as $F_{\mathrm{fuse}}$, are then subjected to processing through the application of two 3×3 depthwise convolutional layers (DW Conv). This is followed by the utilization of a Convolutional Block Attention Module (CBAM), as detailed in the reference [32], to enhance channel and global spatial information. Subsequently, two 1×1 convolutional layers determine the attention weights for the defect segmentation features $F_{\mathrm{seg}}$ and the boundary-aware features $F_{\mathrm{edge}}$. The aforementioned process can be summarized as follows:(4)$$F_{\mathrm{fuse}}={\mathrm{Conv}}_{1\times 1}\left(\mathrm{Concat}\left(F_{\mathrm{seg}},F_{\mathrm{edge}}\right)\right)$$
(5)$$F_{\mathrm{fuse}}^{\prime }=\mathrm{CBAM}\left({\mathrm{DWConv}}_{3\times 3}\left({\mathrm{DWConv}}_{3\times 3}\left(F_{\mathrm{fuse}}\right)\right)\right)$$
(6)$$\begin{array}{cc} F_{\mathrm{seg}}^{\prime } & =\sigma \left({\mathrm{Conv}}_{1\times 1}\left(F_{\mathrm{fuse}}^{\prime }\right)\right)⊙F_{\mathrm{seg}}\\ F_{\mathrm{edge}}^{\prime } & =\sigma \left({\mathrm{Conv}}_{1\times 1}\left(F_{\mathrm{fuse}}^{\prime }\right)\right)⊙F_{\mathrm{edge}} \end{array}$$
where $F_{\mathrm{seg}}$ represents the segmentation decoder features, $F_{\mathrm{edge}}$ represents the boundary-aware features, Concat denotes concatenation, ${\mathrm{Conv}}_{1\times 1}$ is a 1×1 convolutional layer, ${\mathrm{DWConv}}_{3\times 3}$ is a 3×3 depthwise convolutional layer, CBAM stands for the convolutional block attention module, σ represents the sigmoid activation function, and ⊙ denotes the Hadamard product.

Finally, the reweighted features $F_{\mathrm{seg}}^{\prime }$ and $F_{\mathrm{edge}}^{\prime }$ are each processed through a 1×1 convolution and then added to the features from each branch, serving as input to the next layer of the decoder.

#### 2.4.3. Dual-Stream Decoder

Previous research [33] has demonstrated that sufficient boundary information is retained exclusively in low-level features. Consequently, the output features from the third encoder are provided as input to the boundary prediction decoder. The defect segmentation decoder comprises five stages, each of which contains an upsampling block. Subsequently, a 1×1 convolutional layer is applied in order to obtain the final void defect segmentation result. In the initial two stages, elementary addition operations facilitate the interconnection between the encoder and decoder. In contrast, the final three stages encompass the integration of features from the boundary prediction decoder and the defect segmentation decoder, achieved through the implementation of three FCCG modules. This integration ultimately culminates in the formation of our interactive dual-stream decoder as illustrated in Figure 4.

#### 2.4.4. Loss Function

Since our network is based on a dual-stream decoder framework, we employ two loss functions to train the model. For the boundary prediction branch, we use the conventional Binary Cross-Entropy (BCE) loss, defined as follows:(7)$$L_{\mathrm{boundary}}=-\sum_{i}\left[B_{i}^{\det}\log\left(B_{i}^{\mathrm{pre}}\right)+\left(1-B_{i}^{\det}\right)\log\left(1-B_{i}^{\mathrm{pre}}\right)\right]$$
where $B_{i}^{\det}$ and $B_{i}^{\mathrm{pre}}$ represent the predicted boundary map and the ground truth boundary map of the *i*-th image, respectively.

For the defect segmentation branch, we introduce the pixel-position-aware loss [34] function, defined as follows:(8)$$L_{\mathrm{seg}}=L_{\mathrm{wIoU}}+L_{\mathrm{wBCE}}$$
where the weighted IoU loss, $L_{\mathrm{wIoU}}$, addresses the limitations of BCE by considering the overlap between the predicted and ground truth segmentation masks. It is defined as follows:(9)$$L_{\mathrm{wIoU}}=1-\frac{\sum_{i}C\cdot \left(p_{i}\cdot g_{i}\right)}{\sum_{i}w_{i}\cdot \left(p_{i}+g_{i}-p_{i}\cdot g_{i}\right)}$$
where $p_{i}$ and $g_{i}$ represent the predicted and ground truth values for pixel *i*, respectively, and $g_{i}$ is the weight assigned to each pixel based on its importance.

The weighted BCE loss, $L_{\mathrm{wBCE}}$, retains the local pixel-level loss calculation but introduces weights to assign higher importance to specific areas of the image. It is defined as follows:(10)$$L_{\mathrm{wBCE}}=-\frac{1}{N}\sum_{i}w_{i}\cdot \left(g_{i}\cdot \log\left(p_{i}\right)+\left(1-g_{i}\right)\cdot \log\left(1-p_{i}\right)\right)$$
where *N* is the total number of pixels and $w_{i}$ adjusts the contribution of each pixel to the overall loss. By combining these two loss functions, the segmentation model benefits from both global context and local precision, thereby achieving better performance in chip void defect segmentation.

Finally, the overall loss function is formulated as follows:(11)$$L_{\mathrm{total}}=L_{\mathrm{boundary}}+L_{\mathrm{seg}}$$

## 3. Results

### 3.1. Implementation Details

Our deep learning network is implemented using the PyTorch 2.1 framework and the Python programming language. Training is conducted on a high-performance workstation with the configuration detailed in Table 2. During the training phase, the AdamW optimizer is employed, which combines the adaptive capabilities of Adam with weight decay, thereby facilitating more effective control of model complexity and the prevention of overfitting. The initial learning rate is set to $1\times 10^{-3}$, with a weight decay coefficient of $1\times 10^{-4}$. The network is trained for a total of 300 epochs.

### 3.2. Evaluation Metrics

In the experiments, the most widely adopted evaluation metrics in semantic segmentation were employed, namely mean Intersection over Union (mIoU) and Intersection over Union (IoU) for the defect category. These were used as the primary indicators to assess model performance. The definitions of the mIoU and defect IoU evaluation metrics are provided below:(12)$$\mathrm{mIoU}=\frac{1}{2}\left(\frac{TP}{TP+FP+FN}+\frac{TN}{TN+FN+FP}\right)$$
(13)$$\mathrm{Defect}\;\mathrm{IoU}=\frac{TP}{TP+FP+FN}$$
where *TP* represents the number of defective pixels correctly identified and *TN* represents the number of pixels correctly classified as non-defective. *FP* refers to non-defective pixels that were incorrectly identified as defective and *FN* represents actual defective pixels that were incorrectly classified as non-defective.

In comparison to standard mask IoU measurements, boundary IoU is more susceptible to errors pertaining to the boundaries. Accordingly, boundary IoU is included as an evaluation metric for RSALite-UNet. Boundary IoU is defined as follows:(14)$$\mathrm{Boundary}\;\mathrm{IoU}\left(G,P\right)=\frac{|\left(G_{d}\cap G\right)\cap \left(P_{d}\cap P\right)|}{|\left(G_{d}\cap G\right)\cup \left(P_{d}\cap P\right)|}$$
where the notation $G_{d}$ represents the set of ground truth mask contours with a pixel width of *d*, whereas $P_{d}$ represents the set of predicted mask contours with a pixel width of *d*. In the present study, the value of *d* has been set at four pixels.

Two key metrics have been introduced to assess the efficiency of the model. The first metric is called “Params”, which represents the total number of parameters within the model and is used to evaluate the size and complexity of the model. The second metric is GFLOPS, which represents the number of billion floating point operations per second and is used to measure the computational complexity and processing power of the model.

Overall framework evaluation metrics used are accuracy, precision, and recall, defined as follows:(15)$$\mathrm{Accuracy}=\frac{TP+TN}{TP+TN+FP+FN}$$
(16)$$\mathrm{Precision}=\frac{TP}{TP+FP}$$
(17)$$\mathrm{Recall}=\frac{TP}{TP+FN}$$
where *TP* is the number of correctly predicted unqualified chips, *TN* is the number of correctly predicted qualified chips, *FP* is the number of qualified chips falsely predicted as unqualified chips, and *FN* is the number of unqualified chips falsely predicted as qualified chips.

### 3.3. Performance Evaluation of RSALite-UNet

**Performance Comparison:** The proposed RSALite-UNet was evaluated against a number of other networks, including U-Net [19], U-Net++ [35], Attention U-Net [36], DeepLabv3+ [23], and SegFormer [37]. These evaluations were conducted using a custom dataset, CIDX-Ray1950. To ensure that the results were comparable, all networks were tested under identical experimental conditions. The qualitative results are presented in Figure 6. Given the relatively simple topology of the chip-covered regions, all segmentation networks demonstrated efficacy in segmenting images with clear boundaries, as evidenced by the first row of images. However, in images with blurred boundaries, as illustrated in the second row, only RSALite-UNet accurately delineated the chip-covered regions, while the other networks were unable to do so. The proposed RSALite-UNet exhibited consistently superior segmentation accuracy across a range of conditions.

Table 3 presents a quantitative comparison of the segmentation performance of the aforementioned methods, with the most favorable results highlighted in bold. As demonstrated in Table 3, RSALite-UNet attained the highest mean IoU and boundary IoU, reaching 99.52% and 81.27%, respectively. Furthermore, in comparison to other models, RSALite-UNet markedly reduced the model size and computational overhead, necessitating only 4.10 GFLOPs and 3.77 million parameters, whereas the second-ranked U-Net++ required 200.79 GFLOPs and 11.81 million parameters. These findings suggest that RSALite-UNet is capable of achieving high levels of segmentation accuracy while simultaneously reducing the complexity of the underlying network architecture, making it a highly suitable choice for deployment in real-world industrial settings.

### 3.4. Performance Evaluation of DM-UNet

**Performance Comparison:** This section presents an evaluation of the overall performance of the proposed DM-UNet in the context of chip void defect segmentation. The evaluation is conducted by comparing the proposed DM-UNet with other models, including U-Net, DeepLabV3+, SegFormer, VM-UNet, and two U-Net-based models, U-Net++ and Attention U-Net. All models were trained and tested on the self-developed CIDX-Ray1950 X-ray image dataset. The quantitative comparison results of these methods are presented in Table 4, with the optimal results in each column highlighted in bold. As demonstrated in Table 4, the proposed DM-UNet attained the highest scores across all evaluation metrics. Models designed for natural image segmentation tasks, such as DeepLabV3+ and SegFormer, demonstrated suboptimal performance in the internal defect segmentation of chip X-ray images, with boundary IoU scores ranking last (58.60%) and second to last (64.98%), respectively. The U-Net++ model, which enhances the U-Net architecture with DenseNet-based skip connections, enables dense connections at all feature levels, facilitating the integration of multi-scale features and aiding in the extraction of multi-scale defect targets for more precise segmentation. The model achieved a second-place ranking in terms of boundary IoU, with a score of 70.71%. However, the incorporation of these dense connections resulted in a notable reduction in the speed of the segmentation process, which was the slowest among all the models in the GFLOPS metric (200.79). The results demonstrate that the proposed model exhibits enhanced precision in the segmentation of chip packaging defects, achieving an optimal equilibrium between segmentation accuracy and processing speed.

**Ablation Studies:** In order to validate the effectiveness of each improvement in the proposed DM-UNet, the original U-Net was used as the baseline. The quantitative results of the ablation experiments are presented in Table 5, where A represents U-Net, B represents the model using VSS blocks as the encoder layers, and C represents our proposed DM-UNet. As demonstrated in Table 5, the replacement of convolutional blocks in the encoding layers with VSS modules resulted in a decline in the mean IoU, defect IoU, and boundary IoU by 0.18%, 0.88%, and 0.88%, respectively. Despite a decline in the network’s performance, there was a notable reduction in the number of parameters, amounting to 51.83%. This resulted in a considerable enhancement in the inference speed, while the impact on segmentation performance was relatively minimal. Following the introduction of the FCCG module, there was a notable increase in the mean IoU (0.74%), defect IoU (2.35%), and boundary IoU (5.02%), while the computational cost remained relatively low. The results demonstrate that the fusion of features from the two decoder branches can effectively enhance boundary segmentation performance in void defect regions.

Figure 7 shows the segmentation results of DM-UNet on the custom-built CIDX-Ray1950 dataset. The first row displays the original images of the chip cover, the second row shows the annotated chip defect results, the third row presents the binary images from the BGM-UNet network test results, and the fourth row shows the boundary images from the BGM-UNet network test results overlaid with the annotated defects. Green represents the true defect boundaries while red represents the predicted defect boundaries. As shown in Figure 7, the segmentation results of the proposed model more accurately detail defect edges and can identify smaller, harder-to-distinguish defects.

### 3.5. Performance Evaluation of Overall Framework

To comprehensively evaluate the applicability of the proposed chip packaging void defect segmentation framework in real-world industrial scenarios, we used 600 raw X-ray images from the CIDX-1950 dataset as our test set. The test set includes 300 images of qualified chips and 300 images of defective chips, all annotated by professional quality control inspectors.

As shown in Figure 8, the green and blue rectangles represent the outer and inner contours of the welding area, respectively. Red indicates defect edges, yellow represents the width of the annular welding area, and purple denotes the width of the defects. The presence of defects can cause discontinuities in the welding area, potentially leading to the failure of the protective cover’s sealing function, which may result in the cover detaching. In practical industrial applications, a chip is considered defective if the defect width exceeds 50% of the welding area width, or if the total defect area exceeds 5% of the welding area.

The test results are shown in Table 6, where NG represents unqualified chips while OK represents qualified chips. The testing accuracy is 93.3%, and the recall rate is 99.3%, indicating that the proposed void defect segmentation framework is highly reliable. The probability of incorrectly classifying a defective chip as non-defective is extremely low, ensuring that only high-quality chips proceed to the next production stage. In practical applications, since most chips are non-defective, the number of chips falsely classified as defective is minimal. This significantly reduces the workload of quality control personnel in re-inspection tasks. Additionally, on our experimental platform, the average detection time per X-ray image is only 1.6 seconds. Based on these results, the proposed void defect segmentation framework efficiently and accurately identifies and segments void defects in chip packaging.

## 4. Conclusions

In this paper, we propose a deep-learning-based defect segmentation framework specifically designed to detect internal void defects in packaged chips. We first design a lightweight RSALite-UNet network to segment chip covers with different rotation angles. The network efficiently and accurately segments the chip region while reducing background interference in the segmentation of chip void defects. Furthermore, the interactive dual-stream decoder network model (DM-UNet) proposed in this paper, which uses VSS modules as feature extraction layers, better handles the challenges of chip void defects with varying sizes, shapes, and boundary blurs compared to previous semantic segmentation models, while achieving satisfactory segmentation speed. In real industrial scenarios, the proposed framework achieved a precision of 93.3% and a recall rate of 99.3% in detecting chip package void defects. In the future, we will explore semi-supervised methods to address the challenges of void defect labeling and extend the application of intelligent defect detection techniques to a wider range of chip types.

## Figures and Tables

**Figure 1 sensors-24-05837-f001:**
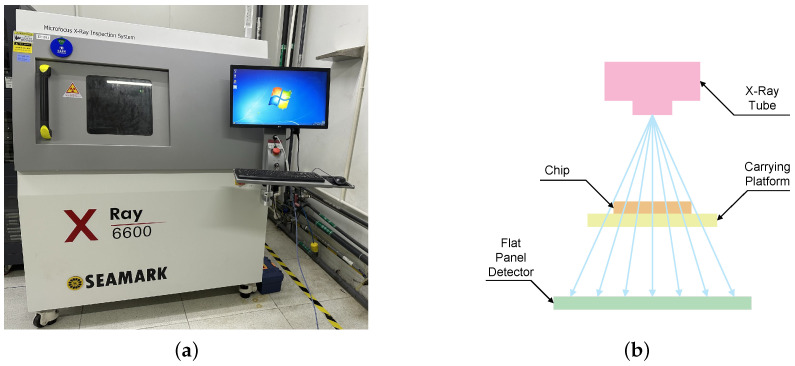
(**a**) X-ray inspection machine. (**b**) The schematic diagram of X-ray image acquisition principle.

**Figure 2 sensors-24-05837-f002:**
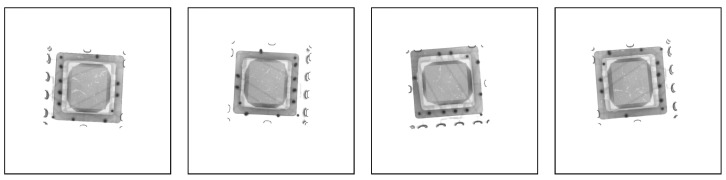
The raw X-ray images obtained by X-ray inspection machine.

**Figure 3 sensors-24-05837-f003:**
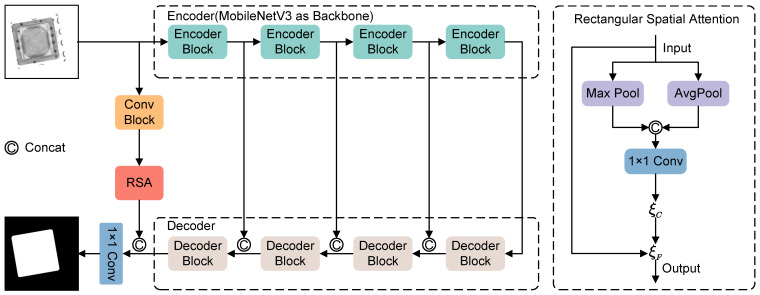
The overall architecture of RSALite-UNet.

**Figure 4 sensors-24-05837-f004:**
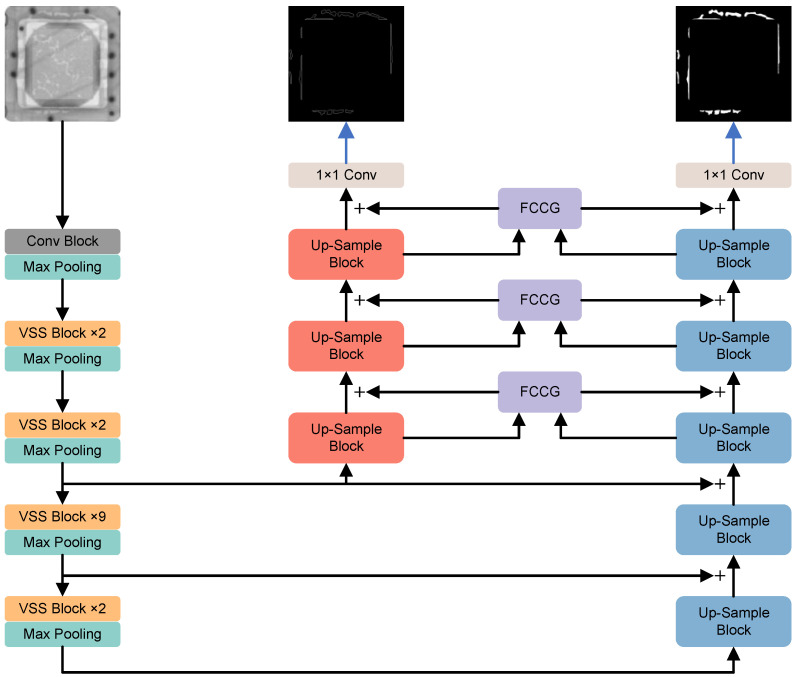
The architecture of our proposed DM-UNet.

**Figure 5 sensors-24-05837-f005:**
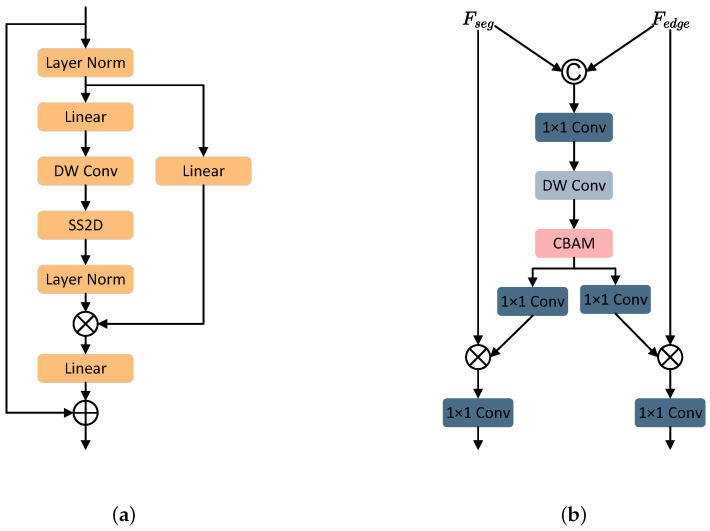
(**a**) VSS Block. (**b**) FCCG Module.

**Figure 6 sensors-24-05837-f006:**
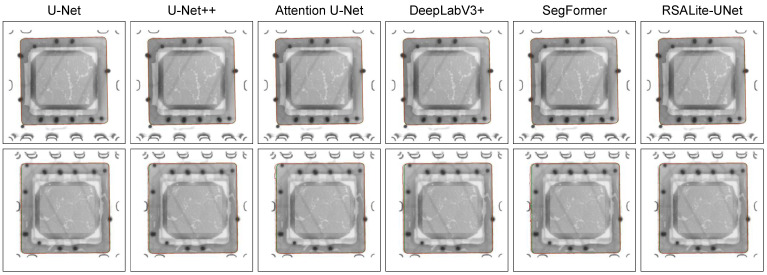
Visualization results of chip cover plate segmentation.

**Figure 7 sensors-24-05837-f007:**
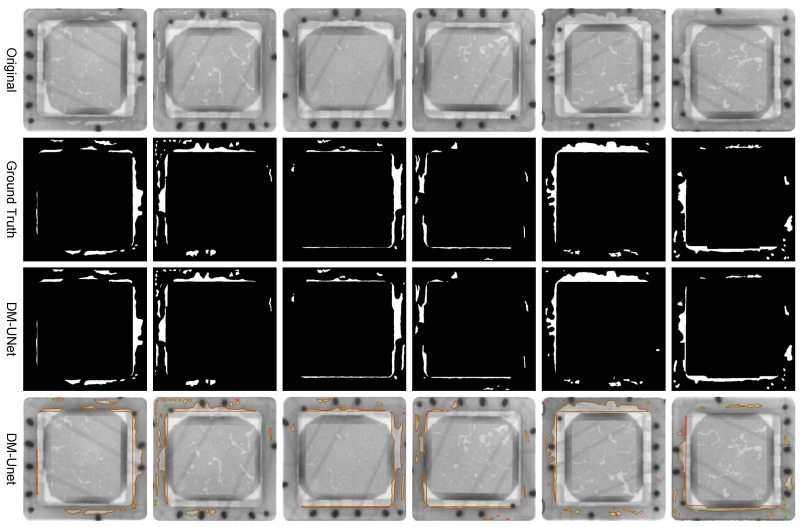
Visualization results of chip void defect segmentation.

**Figure 8 sensors-24-05837-f008:**
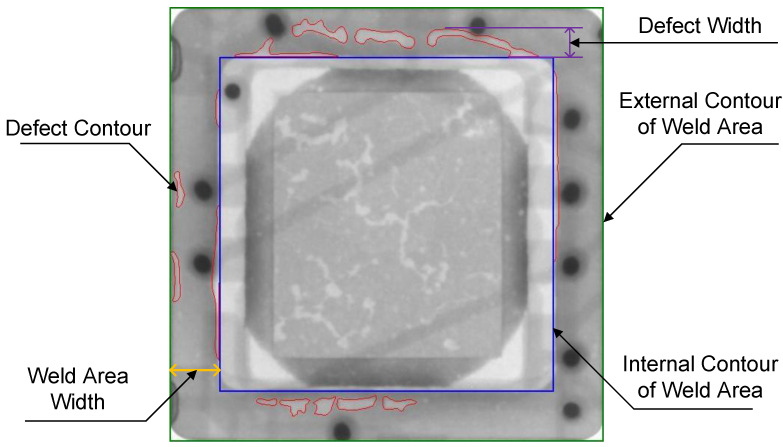
The schematic of chip qualification judgment.

**Table 1 sensors-24-05837-t001:** The distribution of the CIDX-Ray1950 dataset.

	Chip Cover Plate Datasets	Chip Void Defect Datasets	Overall Framework Test Datasets
NG	1024	1024 *	300
OK	326	0	300

NG represents unqualified chip, while OK represents qualified chip. The 1024 * images in the “Chip Void Defect Datasets” column represent a subset of the 1024 NG images from the “Chip Cover Plate Datasets”. These images are specifically annotated for void defects. No additional images were added.

**Table 2 sensors-24-05837-t002:** Workstation configuration.

Component	Specification
CPU	Intel Xeon Gold 5318Y
Memory	256 GB DDR4
GPU	NVIDIA RTX 4090 (24 GB)
Operating System	Ubuntu 22.04 LTS

**Table 3 sensors-24-05837-t003:** Quantitative comparison of different methods for chip cover plate segmentation.

Network	mIoU	Boundary IoU	GFLOPs	Param
U-Net [19]	99.45	78.76	40.51	4.318
U-Net++ [35]	99.46	78.89	200.79	11.81
Attention U-Net [36]	99.42	77.73	66.96	8.72
DeepLabV3+ [23]	99.45	78.29	164.12	39.63
SegFormer [37]	99.40	76.61	71.348	47.22
RSALite-UNet (Ours)	**99.52**	**81.27**	**4.10**	**3.77**

Bold values represent the highest performance indicators.

**Table 4 sensors-24-05837-t004:** Quantitative comparison of different methods for chip void defect segmentation.

Network	mIoU	Defect IoU	Boundary IoU	GFLOPS	Param
U-Net [19]	91.64	85.26	69.11	40.51	4.318
U-Net++ [35]	91.57	85.24	70.71	200.79	11.81
Attention U-Net [36]	91.52	85.13	70.52	66.96	8.72
DeepLabV3+ [23]	90.93	84.10	58.60	164.12	39.63
SegFormer [37]	91.03	83.33	64.98	71.348	47.22
VM-UNet [38]	91.46	84.98	69.20	16.45	22.04
DM-UNet (Ours)	**92.20**	**87.27**	**74.13**	**4.22**	**2.16**

Bold values represent the highest performance indicators.

**Table 5 sensors-24-05837-t005:** Quantitative results of the ablation experiments.

Network	Mamba Encoder	FCCG	mIoU	Defect IoU	Boundary IoU	GFLOPs	Param
A			91.64	85.26	69.11	40.51	4.318
B	✓		91.46	84.92	68.23	3.13	2.08
C	✓	✓	92.20	87.27	74.13	4.22	2.16

**Table 6 sensors-24-05837-t006:** Chip package quality test results.

	Ground Truth		Accuracy	Precision	Recall
	**NG**	**OK**			
**NG**	281	87	93.3%	88.7%	99.3%
**OK**	19	213			

## Data Availability

The data are not publicly available due to the privacy of the dataset.

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
