# Peer review of "A Novel End-to-End Deep Learning Framework for Chip Packaging Defect Detection"

_sensors, 2024, doi:10.3390/s24175837_

Round 1

Reviewer 1 Report

Comments and Suggestions for Authors

The authors proposed a deep learning framework for void defect segmentation in chip packaging. However, there are still some issues that suggest the authors to optimize.

1. The sentence “The source code will be available at https://github.com/Zhousiyi0107/Chip-packaging-defect-detection.git.” is not recommended to appear in the abstract and is suggested to be placed at the end of the introduction;

2. As mentioned in section 2.1, the X-ray tube and FDP can be rotated simultaneously, and the e X-ray inspection system captures images at a fixed resolution. Combined with Figure 2, there should be background pixels outside the Chip. Did the authors cut these pixels? The current description does not match Figure 2. Suggest providing a detailed description in the manuscript;

3. The overlapping parts of the dataset quantity in Table 1 are difficult for readers to understand without description. It is suggested to adjust the structure of Table 1 to facilitate readers' understanding.

4. It is suggested that the authors provide the full name (abbreviation) in the format of the first appearance of "RSAite-UNet" to improve the standardization of the article.

Author Response

Thank you very much for giving us the opportunity to revise our manuscript. We truly appreciate the constructive comments and suggestions provided by the reviewers, which have greatly helped us improve the quality of our paper.

Please refer to the attachment for our comprehensive responses to your comments.

Reviewer 2 Report

Comments and Suggestions for Authors

In the paper the authors presented a deep learning-based defect segmentation framework designed to detect internal void defects in packaged chips. First, a lightweight RSALite-UNet network to segment chip covers with different rotation angles was design, second, the interactive dual-stream decoder network model (DM-UNet) was proposed. Thanks to the use of VSS modules, the model performed better in detecting void defects compared to other semantic segmentation models.

I like the structure of the work, the research has been planned and carried out in a well-considered manner. Abstract, introduction and literature are relevant.

However, I found some parts, which can be misleading:

1. The number of correctly predicted unqualified chips cannot be simultaneously reffered to as TP and TN.

2. The number of unqualified chips falsely predicted as qualified chips cannot be simultaneously reffered to as FP and FN.

3. The content of Table 1 should be discussed above the table; NG and OK explained in the main body of the manuscript. I would repeat it while discussing the Table 6.

4. There is no Table II in the paper.

5. Table 5 - A, B, C should be discussed in the main body of the manuscript.

Comments on the Quality of English Language

I have no significant comments on the English language.

Author Response

(The authors gave the same response as above.)
